# Maybe, Maybe Not: A Survey on Uncertainty in Visualization

Krisha Mehta*

## ABSTRACT

Understanding and evaluating uncertainty play a key role in decision-making. When a viewer studies a visualization that demands inference, it is necessary that uncertainty is portrayed in it. This paper showcases the importance of representing uncertainty in visualizations. It provides an overview of uncertainty visualization and the challenges authors and viewers face when working with such charts. I divide the visualization pipeline into four parts, namely data collection, preprocessing, visualization, and inference, to evaluate how uncertainty impacts them. Next, I investigate the authors' methodologies to process and design uncertainty. Finally, I contribute by exploring future paths for uncertainty visualization.

**Index Terms:** Uncertainty, Data Visualization

## 1 INTRODUCTION

With a rise in complexity and dimensionality of data, analyzing and modeling data becomes more challenging. When most of our decisions are data-driven, it becomes imperative that we know the nature of the data and the patterns it contains. As a result, analyzing the inherent uncertainty in the data is gaining more significance. In various fields, uncertainty can signify different things. For instance, data bias, random or systematic error, and statistical variance are all factors that contribute to data uncertainty. Without understanding the underlying uncertainty in our data, we cannot make accurate predictions. Similarly, to observe the true structure of our data and as well as identify patterns in it, we need to visualize it. Today, we can no longer undermine the significance of uncertainty nor ignore the importance of visualizations for data analysis.

As mentioned before uncertainty is bound to exist whenever there is data. Therefore representation of uncertainty in data visualizations is crucial. Consider the example of hurricane path maps, as shown in Figure 1. The increase in the width of the predicted path with time is not due to an increase in the size of the hurricane. Instead, it is due to representing the inherent uncertainty in the data. In other words, the visualization indicates that compared to Friday, Sunday's hurricane path is more difficult to predict with any degree of accuracy.

Information tends to be withheld from the viewer when one does not portray uncertainty in the visualization. Therefore the viewer might occasionally be ignorant of this exclusion. This breach of trust can have significant consequences for both the author and the viewer. Given this significance, it is reasonable to assume that visualizations frequently include uncertainty. But how often do we encounter charts that represent uncertainty? How frequently do we check for bias in graphs that represent public surveys? As it turns out, not frequently..

In a recent study [9], 121 journalism articles, social science surveys, and economic estimates were examined. Out of 449 visualizations created for inference, the study demonstrates that only 14 accurately depict uncertainty. "What's Going on in This Graph?"

*e-mail: krisha1204@gmail.com

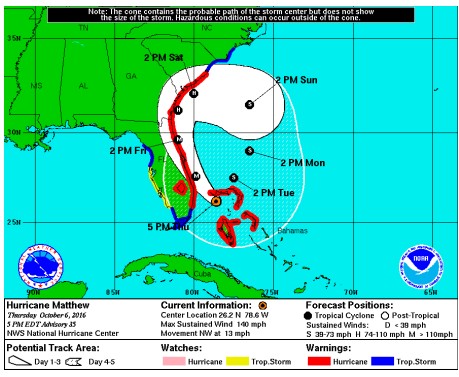

Figure 1: An example chart of a chart for Mattew showing its five-day forecast track [5]

is a New York Times (NYT) initiative to increase graphical literacy, especially among students. Different categories of charts, such as maps, parts-to-whole, and associations, are published for students to explore and analyze. When I looked into the distribution of these charts, I found that only 6 out of the 136 charts show uncertainty.

The question I ask is, do we actually examine uncertainty representations when we come across them in order to make decisions, or do we simply ignore them? Does uncertainty offer value or just clutter these visualizations? I try to investigate these questions in this paper. Visualizations are an integral part of newspapers, government bills, and business earnings reports to name a few. The public uses them to gain insights, spot trends, and make decisions.

Hence, when we visualize data, it becomes critical to support those visualizations with information about uncertainty. People frequently use visualizations to examine data and make observations. Lack of uncertainty representation could result in incorrect and erroneous interpretation. However, it can be challenging to visualize uncertainty. There are no standard guidelines or protocols authors can follow when they create such charts. Given these drawbacks, uncertainty visualization is considered one of the top research problems in data visualization [13]. With the help of a few uncertainty visualization examples and xkcd comic strips, this survey studies how uncertainty contributes to every phase in visualization. Most research in this area focuses on creating charts with uncertainty and how viewers may perceive them. However, uncertainty is also influential in the other parts of the data visualization process, such as during data collection and preprocessing.

**The objectives of this paper are as follows:**

- Provide an entry point for anyone who wants to learn about uncertainty visualization

- Delineate the significance of uncertainty visualizations

- Explore how uncertainty influences every phase of the data visualization process

- Understand the challenges authors and viewers face when interacting with it

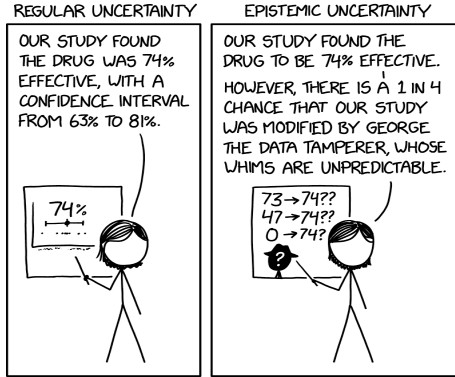

Figure 2: Epistemic Uncertainty [16]

- Discuss the open problems and future research directions in the field

This work is divided into the following sections. Section 2 defines uncertainty and describes the relationship between uncertainty and visualization. In Section 3, I classify the data visualization pipeline into four phases, analyzing the involvement of uncertainty in each phase. Our classification helps look at each phase individually, focusing on the challenges and bottlenecks authors and viewers face when working with uncertainty visualization. Finally, I study some state-of-the-art methods to visualize uncertainty and discuss future directions for research. I conclude the paper in Section 4.

## 2 Uncertainty and Visualization

Visualizations are incredibly important for examining, analyzing, and interpreting data in the era of big data. Visualizations are evidence that a picture really does say a thousand words. They aid viewers in seeing trends, background noise, and outliers. Asking the correct questions can be quite challenging when there is an abundance of data. Through visualizations, viewers can determine what questions the data can help answer. With improvements in hardware, software, and graphics theory, data visualizations are adopted more frequently and widely [29]. Viewers use visualizations to make decisions. However, making decisions and drawing observations by looking at visualizations can be complex due to statistical variance and uncertainty present in these visualizations.

As mentioned previously, uncertainty can have different definitions based on different scenarios [3]. Broadly speaking, uncertainty is classified into two types, aleatory and epistemic. Aleatory uncertainty rises from random fluctuation and unknown outcomes when an experiment is run multiple times in a consistent environment. For example, in a drug trial, a participant's blood pressure can vary due to stress and anxiety. There might also be measurement errors in the sphygmomanometer. Aleatory uncertainty can be minimized by controlling individual factors and increasing the number of readings. Epistemic uncertainty, on the other hand, rises from a lack of knowledge, like predicting the outcome of the same experiment in a completely different, unknown environment. For example, predicting the effect of a drug on a new disease. Uncertainty can be measured, like risks but can also be unquantified, like bias. While aleatory uncertainty is more widely represented in the visualizations [28], both types can be represented with distribution graphs.

Uncertainty and visualizations are interweaved, and working with one often requires working with the other. In 1644, Michael Florent van Langren was one of the first researchers to use visualization for statistical analysis [28]. He used a 1D line graph to present the 12

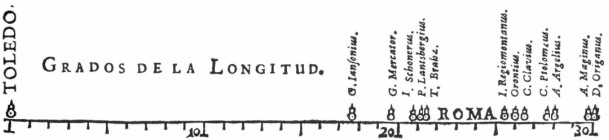

Figure 3: Langren's line graph is one of the first visualizations to present uncertainty

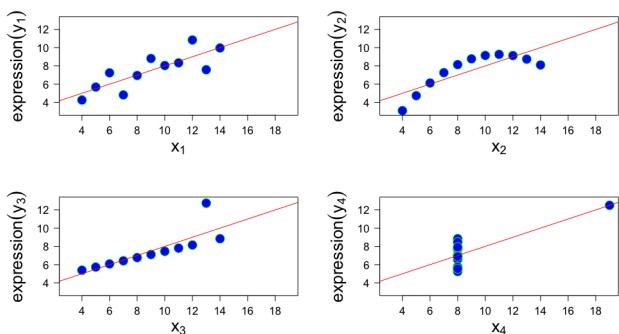

Figure 4: Anscombe's quartet consists for four datasets with similar statistics but very different distributions.

known estimated longitudinal distances between Toledo and Rome, as shown in Figure 3. Instead of using a table to show this data, Langren used this graph to showcase the wide range of variation. Even though all the distances were over-estimated (actual distance, in longitude, is shown using the arrow), the graph remains classic in demonstrating the power of visualization.

The popular Anscombe's quartet [1] is a perfect example of how data with similar statistics might have a very different distribution which is observed when visualized. The quartet consists of four datasets with 11 points having nearly the same mean, sample variance, correlation, linear regression, and coefficient of determination. The four datasets may appear very similar to viewers looking at the data and the descriptive statistics. However, when one visualizes them, the difference in their distribution is very evident, as shown in Figure 4. Looking at data in tabular form may hide insightful observations and can lead to erroneous conclusions. Today, researchers across all domains use extensive libraries such as [4, 11, 12, 22, 25] to analyze data uncertainty.

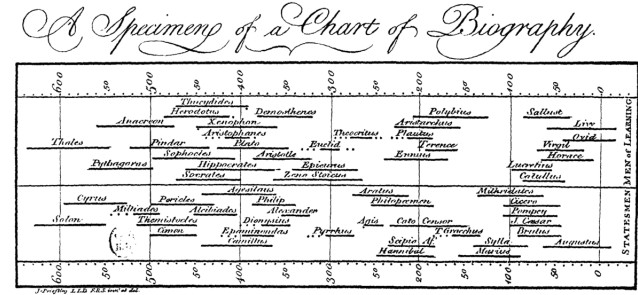

Figure 5: Priestley's Chart of Biography [24]

Using visualizations to represent and study uncertainty in data is widely adopted. However, uncertainty in visualizations is often

not communicated [9]. One of the earliest instances of uncertainty being presented can be traced back to the 18th century. Joseph Priestley, a British scientist, created "A Chart of Biography" to present the lifespans of famous people as shown in Figure 5. He used horizontal lines to portray the lifetime of about 2000 people and used dots before or after the lines to communicate uncertainty.

Visualizations of uncertainty, however, are not common. Numerous factors influence why authors decide against visualizing uncertainty. Since they do not know all the information about the dataset, viewers may draw inaccurate conclusions in the absence of uncertainty representation. Nevertheless, introducing more uncertainty could also make the audience feel too overwhelmed to pay attention to it. The study of why visualizing uncertainty is rare is still in its early stages. In the section that follows, I go through each of these issues in more detail and look at how uncertainty affects every stage of data visualization.

## 3 UNCERTAINTY IN VISUALIZATION

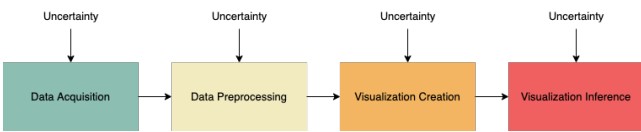

Figure 6: The data visualization process divided into four stages to show how uncertainty affects each stage

Previous works in the field have attempted to classify the data visualization process differently. [14] considers sampling, modeling, visualization, and decision-making as the primary sources of uncertainty. This paper follows a similar classification. I divide the visualization pipeline into **data collection, preprocessing, visualization and inference** as shown in Figure 6 Pang et al. [21] classify the process into data collection, derivation, and visualization and discuss how uncertainty is introduced in each stage.

Under the data collection phase, the paper mainly discusses the uncertainty added due to measurement errors. However, there are other sources, such as bias and sampling error, that the paper fails to describe. I investigate these uncertainties in Section 3.3.1. The authors then discuss the change data undergoes when it is preprocessed. These changes include converting one unit to another, rescaling, and resampling. However, they do not mention other vital issues such as missing data, approximation, and interpolation that I examine in Section 3.3.2. Next, the authors highlight how uncertainty also influences the data visualization stage itself. They mainly focus on radiosity and volume rendering, while our paper delves more into 2D visualizations. Finally, I explore how viewers infer these visualizations and the challenges they face while making a decision from these charts.

Uncertainty is presented at every phase of this classification. However, understanding and evaluating uncertainty in each of these phases is unique. Therefore, authors are required to approach these uncertainties based on their type and complexity, understand their abstraction, and then present them in visualizations in a way that is easy to grasp.

### 3.1 Data Acquisition

Given the interdisciplinary nature of visualizations, the format, quantity, and type of data used to create them vary immensely. Different data implies different data collection processes and uncertainties. Uncertainty is intertwined with data acquisition and can arise from random variables and modeling errors [14].

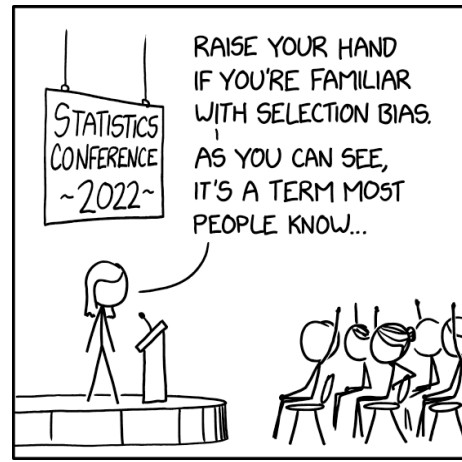

Figure 7: Selection Bias [19]

Pang et al. [21] explain how almost all acquired data has statistical variation. Collected data can have errors, bias, and variance. [26] study how bias can be introduced during the process of collecting data. Datasets are prone to various biases that include but are not limited to selection bias, volunteer bias, admission bias, survivor bias, and misclassification bias.

It is imperative that datasets resemble the true population as closely as possible. Data can also contain different types of errors, such as coverage error, sampling error, nonresponse error, and measurement error [7]. Missing data points is another common challenge researchers face during data collection.

Figure 8: Free Speech, a graph by the New York Times based on a national poll including 1,507 U.S residents [20]

Correcting these errors is not always possible, but they can be mentioned in the visualization to inform the viewer. However, uncertainty is often ignored when authors create visualizations. Other times this uncertainty in data is not communicated to them [9]. For example, when I analyze a piece called "Free Speech" (as shown in Figure 8) published in the What's Going On in This Graph section of the New York Times (NYT) [20], we can see how information about uncertainty from the data source is not mentioned directly in the graph. The bars of the graph do not sum to 100 percent since they are missing the no-response segment. The article mentions that the margin of error for the sample is +/- 3.1%, but the graph makes no mention of it.

Efforts are being made by researchers to improve the way uncertainty in the data collection phase is captured, processed, and

communicated. Athawale et al. [2] propose using statistical summary maps to represent uncertainty in scalar field data caused by data acquisition.

## 3.2 Data Preprocessing

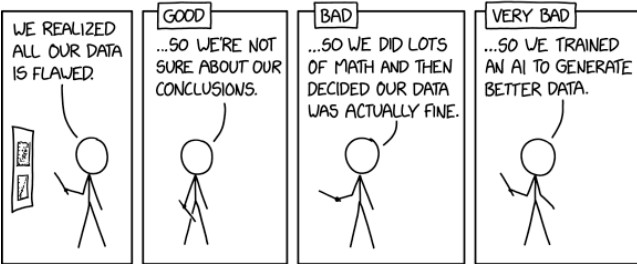

Figure 9: Flawed Data [18]

Raw data is imperfect and can consist of noise and error. Once data is collected, it undergoes processing for accuracy and standardization. However, this phase can add uncertainties to the data that may not be immediately evident. For example, fundamental transformations such as rounding off values, converting data from one unit to another, rescaling, resampling, and quantizing can add uncertainty [1]. Even though this might seem minor, the impact can be significant. For example, based on whether we take the value of pi as 22/7 or 3.159, the area of the Sun can vary by a difference of $237x10^6$ sq. miles.

A significant setback that most datasets suffer from is missing data. Data can have missing values for many reasons, such as instrument malfunction, incomplete observations, and lost data. Missing values leave a gap in the dataset, which makes room for uncertainty. Working with such uncertainty requires the authors to take extra measures during preprocessing. Authors attempt to find close estimates of the missing values to provide the viewers with a complete picture. One way to tackle this problem is by deleting the complete entry that has the missing value. This leads to a loss of data and insights. Another option is to make an educated guess about the missing value. However, this is highly unreliable and often not recommended. Using interpolation, imputation, or other techniques can induce errors [3].

Sometimes, authors choose to encode these estimated values differently in their designs to inform the viewer about the gap in the dataset. However, how authors choose to visualize this encoding becomes very influential in how viewers perceive these graphs. Whether authors highlight, downplay, annotate or remove the missing values determines how much confidence and credibility the viewer shows in the visualization [27].

## 3.3 Visualization Creation

Since uncertainty is ingrained in different parts of the data collection process, it is not easy to identify and control it. However, once the data is cleaned and processed, the authors face a new problem. Creating visualizations is a complicated task that requires authors to make various decisions on behalf of the viewer. Authors are expected to choose the type of visualization based on data type, which may lead them to choose the scaling, sorting, ordering, and aesthetics [30]. Compelling visualizations are accurate and suggest an understanding and interpretation of data. Hence, it is the author's responsibility to analyze data correctly before creating any visualizations. Midway [15] describes ten design principles authors can follow to create charts. However, none of those principles discuss how uncertainty can be presented. Creating effective visualizations is hard. However,

when we add uncertainty representation, the task becomes much more complex. The data visualization community of researchers, designers, journalists, etc., has been reluctant to add uncertainty to their charts. Authors are aware of how significant uncertainty visualization is. Yet, they choose to exclude uncertainty when they design their charts for various reasons discussed below.

### 3.3.1 Uncertainty is hard to represent

Though data is replete with uncertainty, the difficulty lies in determining if it should be represented and how. If the uncertainty has no direct relationship to the goal of the visualization, then it may not be included in the visualization. But this is not a conclusion that authors can quickly draw. The rise in techniques of visualizing uncertainty can make it harder for authors to decide which one to choose from. One of the biggest challenges in visualizing uncertainty is discovering and communicating the relationship and impact that the uncertainty has on the data. Data visualization is often a preferred choice for analysis due to its ability to present high-dimensional data. However, uncertainty also has dimensions, generally classified into scalar, vector, and tensor [23]. While scalar and vector fields of uncertainty are depicted in charts, tensor fields are often avoided. Mapping these dimensions of uncertainty along with the dimensions of data is challenging and often overlooked when creating charts. Instead, authors tend to simplify uncertainty to align with the dimensionality of the data.

### 3.3.2 Uncertainty is hard to calculate and verify

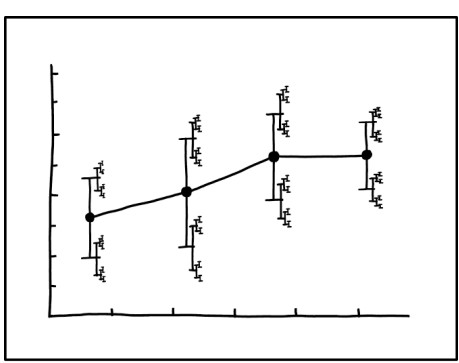

Figure 10: Error Bars [17]

Another reason why authors choose to exclude uncertainty from their charts is that calculating uncertainty is complex [9]. It is well known that even mathematicians and statisticians sometimes find it challenging to calculate the error or variance in a dataset. Verifying if the presented uncertainty is correct is challenging. Moreover, if the authors make an error while designing their charts, they end up providing wrong information to the viewers and losing their trust.

### 3.3.3 Viewers may be overwhelmed

[9] explains why the inclusion of uncertainty in graphs is not widely adopted. Authors believe that uncertainty can be challenging for the viewers to perceive and understand. As a result, viewers may choose to either look at an alternative graph that does not contain any uncertainty representation or overlook the uncertainty in their graph altogether.

### 3.3.4 Uncertainty can add clutter to the visualization

Authors can be unsure of how effective communicating uncertainty is. They also worry about adding more information to an already

visually complex visualization. For many authors, the goal of a chart is to express a signal [9] that can be useful to their viewers. This signal tends to present a single point or a single source of truth. Uncertainty tends to challenge that notion by obfuscating the signal. Additionally, expressing the intricacy of uncertainty through a visual abstraction is challenging. The dimensionality of the data also plays a vital role in deciding whether uncertainty should be represented or not. An increase in the dimensionality of data makes it harder for the human visual system to perceive it effectively. Sometimes even two-dimensional charts can be overwhelming for the viewer. In such a case, representing uncertainty adds visual overload [23].

### 3.4 Visualization Inference

Uncertainty is hard to understand and analyze. When faced with perceiving an uncertain visualization, viewers can get confused or derive inaccurate information from it. One easy method viewers tend to use is to ignore the uncertainty in the graph altogether. Another way is to substitute tricky calculations with easy ones or use heuristics to make decisions. However, this may not always give a correct observation. The most common approach to show uncertainty is by using box plots and error bars. Though widely used, viewers may find them challenging to analyze [6]. Sometimes visualizing uncertainty as frequency instead of distribution provide a better understanding.

Currently, research is being done to create visualizations are help understand uncertainty more intuitively. For example, hypothetical outcome plots (HOPs) represent uncertainty by animating a finite set of individual draws [10]. This approach expects no prior knowledge of the domain from the viewer. However, using HOPs in physical media might be challenging. Bubble treemaps [8] are another approach for visualizing uncertainty. These circular treemaps encode additional information about uncertainty by allocating additional space for visuals.

While uncertainty is still underrepresented in visualizations, more researchers are slowly adding it to their designs. One of the significant setbacks in uncertainty visualizations for authors is calculating uncertainty, while for viewers, it is graphical literacy. Efforts can be taken to increase this literacy through different programs gradually. Furthermore, work should be done to understand what visualization type best suits a given uncertainty type. This relationship can also depend on the type of data being represented and the target audience viewing the graph. For example, it is necessary for graphs published in newspapers and reports to be easily understandable by the public. Hence, studies focusing on visualizing uncertainty with no prior knowledge or information can be very insightful.

### 4 CONCLUSION

Uncertainty visualization is one of the most complex research areas in data visualization today. This work provided an overview of uncertainty visualization and the relationship between uncertainty and visualization. I divided the visualization pipeline into four phases and surveyed papers to study how uncertainty interacts with each phase of the process. The work also investigated why the representation of uncertainty is not widely practiced by the data visualization community and the challenges viewers face when inferring from such a graph. Lastly, I discussed a few state-of-the-art methods to design uncertainty visualization and offered a glance into the interesting future research this field has to offer.

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
