# OpenReview forum: "Maybe, Maybe Not: A Survey on Uncertainty in Visualization"
_IEEE.org/2022/Workshop/altVIS — Reject_

### Official Review · Reviewer_YhZv · 2022-08-09

**Review:**

This paper does a good job of outlining the field of uncertainty visualization and explaining the challenges of the field. It doesn't, however, feel provocative or alternative in any manner. Its only "joke" value comes from the author using xkcd strips to illustrate the points.

**Conflicts:**

None.

**Review Inclusion:**

No

**Sufficiently Alt:**

No

**Superlative:**

The most uncertain.

---

### Official Review · Reviewer_RiLo · 2022-08-19

**Review:**

This paper gives an overview of the history of uncertainty in visualizations (to emphasise on the fact that we still do not use uncertainty enough even though it is an aspect of visualization that has existed for a long time), along with a description of the challenges that arise for authors and viewers during four phases that compose visualization, that are the data collection, the pre-processing of data, the creation of visualization (the choices that are made for the design) and the inference (how viewers read, analyse and understand the visualization).

__PROS:__
1. This paper is well illustrated and it helps understand some notions in a humorous way.
1. The classification used is clear and well-thought so that interesting aspects of the challenges of uncertainty in visualizations can be brought up and explained. For visualization designers (and researchers, etc), this classification might be useful in order to identify how uncertainty could be included in their work and how to best aim for it.

__CONS:__
1. There are some questions that are brought up in the introduction (which are really interesting) such as "do we actually examine uncertainty representations when we come across them in order to make decisions, or do we simply ignore them?", but that are not really addressed in this paper. There are two subsections (3.3.3 and 3.3.4) in which there is an attempt to discuss this question but that seem to be focusing on authors' POV. It could be clearer if this was a part of the 'inference' section with a more in-depth investigation of viewers' attitude towards uncertainty in visualizations.
1. In this paper, it is argued that uncertainty is important in visualization and that "uncertainty visualization is considered one of the top research problems in data visualization". It is indeed a research field on its own and this survey would probably fit in other parts of the conference that address uncertainty. While uncertainty visualization has been 'alt' at some point, I also believe there is now more researchers that are interested and that are investigating this area for this paper to be about 'alternative visualizations'.

__Additional comments:__
1. Decision-making is mentioned and some papers that have studied how participants make decisions when being confronted with uncertainty visualizations could be included or discussed (_Uncertainty Displays Using Quantile Dotplots or CDFs Improve Transit Decision-Making_ by Fernandes et al., 2018, _When (Ish) is My Bus? User-Centered Visualizations of Uncertainty in Everyday, Mobile Predictive Systems_ by Kay et al., 2016, _Decision-Making under Uncertainty: How the Amount of Pre- sented Uncertainty Influences User Behavior_ by Greis et al., 2016).
1. In the introduction, it is written "There are no standard guidelines or protocols authors can follow when they create such charts." and while we are most certainly lacking research in this area, this sentence suggests that no work has been done to help authors design appropriate uncertainty visualization in specific contexts at least.
1. Figures 1 and 3 could be interesting if explained better. It is difficult to understand how the uncertainty is depicted and as such, the figures lose their _illustrating power_. (For the first figure, it is poorly designed enough so that it becomes quite hard to parse, and for the third figure, first it is written that this is the first time visualization has been used for statistical analysis, but the description says "Langren’s line graph is one of the first visualizations to present **uncertainty**").
1. This _claim_ is unclear: "It is imperative that datasets resemble the true population as closely as possible." in 3.1.

__Typos, syntax, etc:__
1. Comma needed in the second part of the first paragraph of the introduction: “As mentioned before, uncertainty […]”.
1. Figure 1: the name of the hurricane is Matt**h**ew and the figure’s description is repetitive (“an example chart of a chart […]”).
1. Paragraph right before second section: “__our__ classification” when everything else is at the first person (same thing in the third section).
1. Figure 4: I am not sure “consists __for__” is the correct way to phrase this.
1. First paragraph of section 3: A period is missing after “as shown in Figure 6”.
1. In 3.4: “Currently, research is being done to create visualizations __are__ help understand uncertainty more intuitively. “

Overall, this paper is good at what it should be doing: the classification helps understand the challenges and the reason(s) why uncertainty should be included more often is well explained. However, there are some parts that could be supplemented with visualization examples that would provide a better overview of how the field is evolving. It is very descriptive and I would have liked to see how researchers and designers, nowadays, have tried to take part of this big 'uncertainty' journey when compared to the beginning of this field.

__Clarity:__ This paper is clear enough thanks to the illustrations although there are a few details (as mentioned earlier) that could be better described (such as some figures). What might still feel a bit unclear is the link between some questions asked at the beginning of the paper and what is actually addressed overall.

__Originality__: A similar paper is referenced but the differences between the paper from Pang et al., and this one are explained.

**Conflicts:**

I have no conflict of interest with the author of this work.

**Review Inclusion:**

No

**Sufficiently Alt:**

No

---

### Official Review · Reviewer_h2am · 2022-08-23

**Review:**

This paper presents an overview of some key terms in uncertainty visualization. A STAR report on uncertainty vis is certainly worth doing, but I question if alt.vis is the right venue for this work.

Uncertainty visualization, while complex, is a pretty mainstream topic in visualization. There has often been at least one or two paper sessions devoted to uncertainty vis papers in the main IEEE VIS conference for the past few years, it is often an entire unit in graduate school visualization courses, is a common topic for journal special issues or workshops, and even surveys of the work in uncertainty visualization are pretty common (the most recent one I could find was from early this year https://www.frontiersin.org/articles/10.3389/fbinf.2022.793819/full). I could see this sort of survey work being sufficiently "alt" if, for instance:
1) The topic was "out there" enough to be alt by itself.
2) The topic was normal enough in other fields, but had been overlooked in VIS, and so is alt from the perspective of our community.
3) The topic was neither "out there" nor overlooked in VIS, but the authors' perspective on the topic was idiosyncratic, challenging, or otherwise provocative enough to be alt.

1) and 2) do not appear to be the case. Ergo, I think 3) is the main viable path here. Plenty of other works have suggested that uncertainty visualization is complicated, that it can arise at multiple stages of the visualization pipeline, and that designers often don't include it (sometimes for good reasons). How is the author specifically adding to this discussion, or troubling it, or otherwise offering an alternative to the status quo?

**Conflicts:**

No conflicts as far as I know,.

**Review Inclusion:**

No

**Sufficiently Alt:**

No

**Superlative:**

Most uncertain

---

### Official Review · Reviewer_txQQ · 2022-08-24

**Review:**

The paper is original in elaborating on uncertainty in visualizations.
Pros:
1. uncertainty vs. visualization and its effect on readers' trust of the visualizations is an important and original topic
2. the author has a comprehensive discussion of the research question on a data science pipeline including data acquisition, data pre-processing, visualization creation, and visualization inference


Cons:
the definition of risk, uncertainty, and other related concepts needs to be discussed in more details. For example, the author should consider the terminology "ambiguity": https://en.wikipedia.org/wiki/Ambiguity


**Conflicts:**

NA

**Review Inclusion:**

Yes

**Sufficiently Alt:**

Yes

**Superlative:**

Most controversial

---

### Official Review · Reviewer_976i · 2022-08-31

**Review:**

In this paper, the authors proposed a survey on uncertainty in visualisation. Reviewers agreed that this is a very interesting topic for the vis community and the paper was humorously illustrated. However, they also felt it was a topic already present at vis and quite discussed, thus maybe not Alt enought for the workshop. We decided to reject the paper for AltVis this year.

**Conflicts:**

No Conflicts

**Review Inclusion:**

No

**Sufficiently Alt:**

No

---

### Decision · Program_Chairs · 2022-08-31

Reject